# Forecasting under-five stunting in Ethiopia using classical and machine learning time series models

Rewina Tilahun Gessese[1]*, Jenberu Mekurianew Kelkay[2], Fetlework Gubena Arage[3], Tigist Kifle Tsegaw[3], Zinabu Bekele Tadese[4], Meron Asmamaw Alemayehu[5], Eliyas Addisu Taye[6], Eyob Akalewold Alemu[3]

**1** International Institute for Primary Health Care –Ethiopia, Addis Ababa, Ethiopia, **2** Department of Public Health, College of Health Sciences, Debark University, Debark, Ethiopia, **3** Department of Public Health, Institute of Public Health, College of medicine and Health Science, University of Gondar, Gondar, Ethiopia, **4** Department of Health Informatics, College of Medicine and Health Sciences, Samara University, Semera, Ethiopia, **5** Department of Epidemiology and Biostatistics, Institute of Public Health, College of Medicine and Health Sciences, University of Gondar, Gondar, Ethiopia, **6** Department of Health Informatics, Institute of Public Health, College of medicine and Health Science, University of Gondar, Gondar, Ethiopia

* rewinatilahun06@gmail.com

## Abstract

### Background

The prevalence of under-five stunting in Ethiopia remains above 30%, which, according to World Health Organization (WHO), constitutes a major public health concern. Stunting has long-term consequences for child growth, development, and overall health. Accurate forecasting of its prevalence is therefore essential to guide policymakers and inform targeted interventions. This study aimed to forecast the prevalence of under-five stunting in Ethiopia for the period 2025–2030 using historical data and time series modeling.

### Methods

Annual under-five stunting prevalence data for Ethiopia from 2000 to 2024 were retrieved from the WHO Global Health Observatory. Time series forecasting models, including Autoregressive Integrated Moving Average (ARIMA), Exponential Smoothing (ETS), Multilayer Perceptron (MLP), and Long Short-Term Memory (LSTM), were developed and evaluated. Model performance was assessed using time series cross-validation with mean absolute error (MAE), mean absolute percentage error (MAPE), and R² as evaluation metrics. The best-performing model was applied to forecast stunting prevalence for 2025–2030.

**Data availability statement:** All relevant data are within the paper and its Supporting Information file.

**Funding:** The author(s) received no specific funding for this work.

**Competing interests:** The authors have declared that no competing interests exist.

## Results

The ETS model demonstrated the best predictive performance (MAE = 1.09, MAPE = 2.71%, R² = 0.903) and was selected for forecasting. Forecasting indicated a gradual decline in under-five stunting prevalence in Ethiopia from 33.95% in 2025 to 31.95% in 2030. The projected prevalence for 2029 is 32.35% (95% CI: 26.90–37.79%), above the national target of 19%, and the 2030 forecast remains well above the SDG target of ending all forms of malnutrition.

## Conclusion and recommendation

Under-five stunting in Ethiopia is projected to remain above national and SDG targets by 2030, indicating current nutrition efforts are insufficient; national program evaluation and evidence-based policy adjustments are recommended.

## Background

Malnutrition refers to deficiencies, excesses, or imbalances in a person's intake of energy and/or nutrients. It encompasses undernutrition (including stunting, wasting, underweight, and micronutrient deficiencies) as well as over nutrition (overweight and obesity) [1]. Among the forms of undernutrition, stunting is defined as a height-for-age Z-score below −2 standard deviations from the World Health Organization (WHO) Child Growth Standards and represents chronic nutritional deficits in children under five [2].

In 2024, about 150.2 million children under five were stunted worldwide, with most cases occurring in Asia (51%) and Africa (43%) [3]. In Ethiopia, the 2019 Ethiopia Mini Demographic and Health Survey reported that 37% of children under five were stunted, while a 2023 nationwide nutrition survey conducted by Ministry of Health and partners estimated the prevalence at 39% [4,5]. More recently, a 2024 systematic review and meta-analysis reported a pooled prevalence of 43% [6]. Overall, these findings show that the prevalence of stunting in Ethiopia ranges between 37% and 43%. All estimates exceed the WHO threshold of 30%, highlighting the very high public health significance of child stunting [2].

Stunting is associated with multiple adverse outcomes in children. Health-related complications include increased susceptibility to infections such as diarrhea, pneumonia, and measles due to impaired immunity, as well as higher child mortality, with 45% of all child deaths linked to undernutrition worldwide [7,8]. Stunted children also experience delayed physical growth, which may result in permanent short stature, and impaired cognitive development, leading to lower learning capacity, poorer school performance, and reduced educational attainment. In addition, stunting increases the risk of chronic diseases such as obesity, and cardiovascular conditions later in life [8–12]. In Ethiopia, 28% of all cases of child mortality are associated with the higher risk of undernutrition. Childhood undernutrition also imposes an economic cost equivalent to 0.5% of the total public health budget, while productivity losses linked to reduced educational attainment and diminished adult earnings are estimated to cost the country up to 16% of its annual GDP in Ethiopia [13,14].

Addressing stunting is central to achieving the Sustainable Development Goals (SDGs), particularly SDG 2 (Zero Hunger) and SDG 3 (Good Health and Well-being). Recognizing this, the Ethiopian government has implemented programs and strategies such as the National Nutrition Program (NNP), the Seqota Declaration, and community-based interventions including growth monitoring and micronutrient supplementation [15–19]. While these initiatives have yielded some gains, persistent socioeconomic, environmental, and behavioral barriers continue to slow progress [6,20,21].

Accurate forecasting of under-five stunting is essential to guide Ethiopia's response. Forecasts enable monitoring of progress toward national and global nutrition targets, improve resource allocation, and support evidence-based policy design. For forecasting, traditional statistical methods, such as ARIMA and Exponential Smoothing (ETS) as well as deep learning machine learning approaches, including MLP, and LSTM networks, offer promising tools for capturing temporal dynamics and generating reliable projections [22–24]. By comparing these approaches, robust forecasts can be generated to inform policymakers, strengthen intervention strategies, and assess whether ongoing efforts are sufficient to meet nutrition targets. Against this background, the current study seeks to evaluate and compare statistical and machine learning models for forecasting under-five stunting in Ethiopia, with the ultimate goal of identifying the most accurate and practical approach for predicting the future burden.

## Method

### Data preparation and preliminary assessment

The analysis was conducted on a univariate time series dataset comprising 24 annual observations of under-five stunting prevalence in Ethiopia, spanning from 2000 to 2024. The data were obtained from the WHO Global Health Observatory repository and extracted programmatically using Python, with the *pandas* and *requests* libraries to facilitate automated retrieval. The raw data were loaded into a pandas DataFrame, with the Year column converted to a datetime object and set as the index to maintain chronological order. Missing data were checked.

Prior to modeling, the dataset underwent a preliminary assessment to ensure readiness for time series forecasting. Line plots were inspected to identify trends or anomalies. Stationarity is a fundamental requirement for many time series models, including ARIMA, because it ensures that the statistical properties of the series such as mean and variance remain constant over time, enabling reliable forecasts [25]. To assess stationarity, the series was first visually inspected using rolling mean and rolling standard deviation over a three-year window. Formal statistical tests were then applied using the Augmented Dickey-Fuller (ADF) test ($p < 0.05$) indicates stationarity). Differencing was applied where non-stationarity was detected, and the tests were repeated to confirm stationarity. Finally, the series was decomposed using STL (Seasonal-Trend decomposition using Loess) to isolate trend, seasonality and residual components (S1 Fig).

For the deep learning models (MLP and LSTM), additional preprocessing was applied to the dataset. The series was scaled to the range [0, 1] using ***MinMaxScaler*** to improve convergence during model training. Input sequences were then constructed using lagged values of the series (the previous three years of prevalence to predict the subsequent year) to serve as features.

### Modeling approaches

To forecast under-five stunting prevalence, multiple candidate models were implemented, including classical statistical models (ARIMA and ETS), deep learning models (MLP and LSTM), and a hybrid ETS–LSTM approach. These models were systematically assessed to identify the best-performing model for forecasting.

### Classical time series methods

ARIMA is a widely used time series model that captures both autoregressive (AR) and moving average (MA) components, with differencing applied to ensure stationarity [26]. The ARIMA model is expressed as ARIMA (p,d,q)), where p represents

the autoregressive order, d the degree of differencing, and q the moving average order. ARIMA is a widely used time series approach that captures both autoregressive (AR) and moving average (MA) components, with differencing applied to ensure stationarity. For this study, the series was first tested for stationarity using the Augmented Dickey–Fuller (ADF) test using the *statsmodels* library. Candidate values for p and were guided by inspection of the autocorrelation (ACF) and partial autocorrelation (PACF) plots, and set as p = 0,1,2,3,5,6 and q = 0,1,2. A systematic grid search was then conducted across these ranges, with model comparison based on the Akaike Information Criterion (AIC) (S2 Fig).

The ETS approach was employed using *statsmodels* as a second classical statistical model. ETS models decompose a time series into three components: Error (E), Trend (T), and Seasonality (S), which can each be specified as either additive (A) or multiplicative (M), or set to none (N) if not present [27]. The ETS framework was applied to capture level and trend components of the annual series. Candidate models considered included ANN (Additive Error, No Trend, No Seasonality), AAN (Additive Error, Additive Trend, No Seasonality), AMN (Additive Error, Multiplicative Trend, No Seasonality), MNN (Multiplicative Error, No Trend, No Seasonality), MAN (Multiplicative Error, Additive Trend, No Seasonality), and MMN (Multiplicative Error, Multiplicative Trend, No Seasonality). Seasonality was excluded based on the results of the STL analysis. A grid search over the level and trend smoothing parameters was conducted, with model selection based on the Akaike Information Criterion (AIC).

## Deep learning models

Deep learning models are a subset of machine learning algorithms capable of automatically learning complex, hierarchical representations from data. They are particularly well-suited for time series forecasting, as they can capture nonlinear patterns, temporal dependencies, and long-term trends that traditional statistical models may fail to detect [28,29]. In this study, two deep learning architectures Multilayer Perceptron (MLP) and Long Short-Term Memory (LSTM) networks were implemented to explore their potential in forecasting under-five stunting prevalence. These models were chosen based on their demonstrated effectiveness in prior studies and their strong performance in capturing temporal patterns in health and epidemiological time series data [30,31]. Both models used the dataset prepared and assessed in the Data Preparation and Preliminary Assessment phase.

MLPs were implemented using scikit-learn. Various architectures were evaluated, including single hidden-layer and shallow two hidden-layer networks with configurations: (5,), (8,), (5,3), (8,4), and (10,5). Activation functions tested included ReLU and tanh. Key hyperparameters and their search ranges were: L2 regularization α (0.001–0.5), learning rate (0.001–0.1), batch size (2–8), and maximum iterations (500–1500).

Hyperparameter tuning was performed using RandomizedSearchCV, allowing efficient exploration of the hyperparameter space. Early stopping with patience between 20–50 epochs was applied to prevent overfitting. The final MLP model selected was a shallow two-layer network with 8 and 4 units in the first and second layers, respectively, ReLU activation, L2 regularization α = 0.1, learning rate 0.1, batch size 8, and a maximum of 1000 iterations. This configuration provided optimal validation performance while minimizing model complexity, which is important given the small dataset.

LSTM networks were implemented using TensorFlow Keras. A single LSTM layer with 2–5 units was followed by a dropout layer (0.2–0.5) and a dense output layer for regression. L2 regularization (l1_l2) was applied to both kernel and recurrent weights. The Adam optimizer was used with mean squared error as the loss function.

Given the higher computational cost and the small size of the dataset, hyperparameter tuning for LSTM was conducted via a controlled manual random search across the following ranges: number of units (2–5), dropout rate (0.2–0.5), L2 regularization strength (0.001–0.1), learning rate (0.001–0.1), batch size (2–8), and number of epochs (50–150). Early stopping (EarlyStopping) and learning rate reduction callbacks were applied to ensure stable convergence and prevent overfitting. **The** final LSTM model used 4 units, dropout rate 0.2, L2 regularization 0.001, batch size 8, 50 epochs, and a learning rate of 0.01. This shallow, single-layer architecture with regularization strategies was chosen to maintain model simplicity and robustness, while still capturing temporal dependencies in the annual stunting data

 

Based on the performance of the statistical model, a hybridization approach with LSTM was evaluated. Residuals from the best-performing statistical model were extracted and assessed for remaining patterns. An LSTM model was then trained on these residuals using the same hyperparameter tuning, dropout, L2 regularization, early stopping, and time-series cross-validation procedures applied to the standalone LSTM model. The predictive performance of the hybrid model was compared with a naive baseline model, which predicts the mean of the residuals, to determine whether the LSTM captured additional meaningful patterns beyond those already modeled by the statistical approach.

### Model performance evaluation

Predictive performance of all models was assessed using time series cross-validation, implemented via the *TimeSeriesSplit* function from *scikit-learn*, ensuring that training data always preceded testing data chronologically. For each fold, models were trained on the training subset and evaluated on the test subset. Performance metrics included mean absolute error (MAE), root mean squared error (RMSE), mean absolute percentage error (MAPE), and coefficient of determination ($R^2$). For deep learning models, early stopping and learning rate reduction were applied during training to prevent overfitting and ensure stable convergence. Model selection was based on cross-validated performance across these metrics.

## Results

### Temporal patterns of under-five stunting in Ethiopia

Over the 24-year period from 2000 to 2024, the prevalence of stunting among children under five in Ethiopia shows a clear declining trend, decreasing from approximately 57% to around 35%. Minor fluctuations observed in certain years reflect short-term variability (Fig 1).

### Stationarity test

Visual inspection of the rolling statistics (rolling mean and standard deviation) of the original under-five stunting series revealed noticeable trends and fluctuations over time, indicating that the series was non-stationary. This observation was formally confirmed using the Augmented Dickey-Fuller (ADF) test, which yielded a p-value of 0.62, supporting the presence of a unit root. To stabilize the series, first-order differencing was applied. Post-differencing, the ADF test indicated stationarity ($p < 0.001$), and the rolling mean and standard deviation plots showed a stable mean and variance throughout the series (Figs 2, 3).

### Modeling approaches for forecasting under-five stunting

**Classical model.** For ARIMA, the series was first differenced to achieve stationarity. After evaluating candidate autoregressive and moving average orders guided by autocorrelation and partial autocorrelation patterns, ARIMA (1,1,2) was selected as the best-fitting model based on the lowest AIC. This model demonstrated predictive performance of MAE = 1.15, MAPE = 2.82%, and $R^2$ = 0.88. Among the candidate ETS models, the AAN (Additive Error, Additive Trend, No Seasonality) specification had the lowest AIC (20.00). This ETS (A,A,N) model demonstrated predictive performance with MAE = 1.09, MAPE = 2.71%, and $R^2$ = 0.90.

**Deep learning model.** For the multilayer perceptron (MLP) model, the final input feature set after hyperparameter tuning included prevalence, lag_1, lag_2, lag_3, rolling_mean_2, rolling_std_2, rolling_mean_3, rolling_std_3, and time_index. The model was selected based on the lowest cross-validation MAE of 0.426. The optimized network consisted of two hidden layers with 8 and 4 neurons, respectively, using the ReLU activation function. Other selected hyperparameters were: n_iter_no_change = 20, max_iter = 1000, learning_rate_init = 0.1, early_stopping = True, batch_size = 8, and alpha = 0.1. Predictive performance metrics were MAE = 0.426, MAPE = 7.14%, and $R^2$ = 0.49% LSTM model achieved

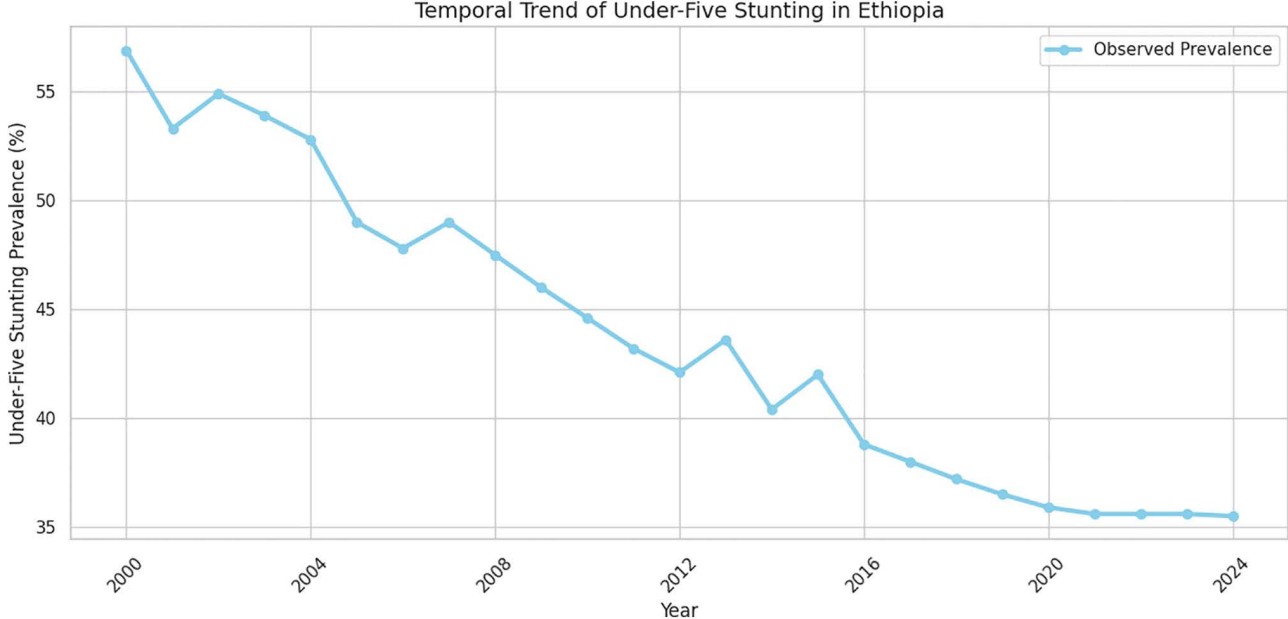

**Fig 1. Temporal Trend of Under -Five stunting in Ethiopia.**

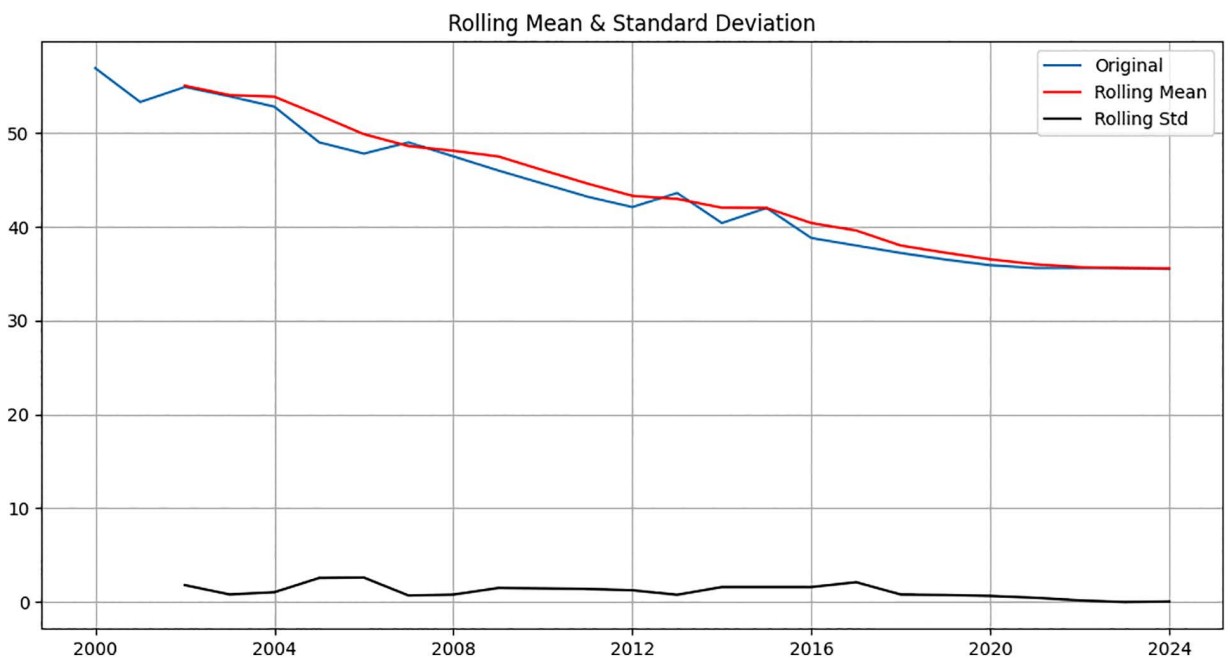

**Fig 2. Rolling mean and SE before differencing.**

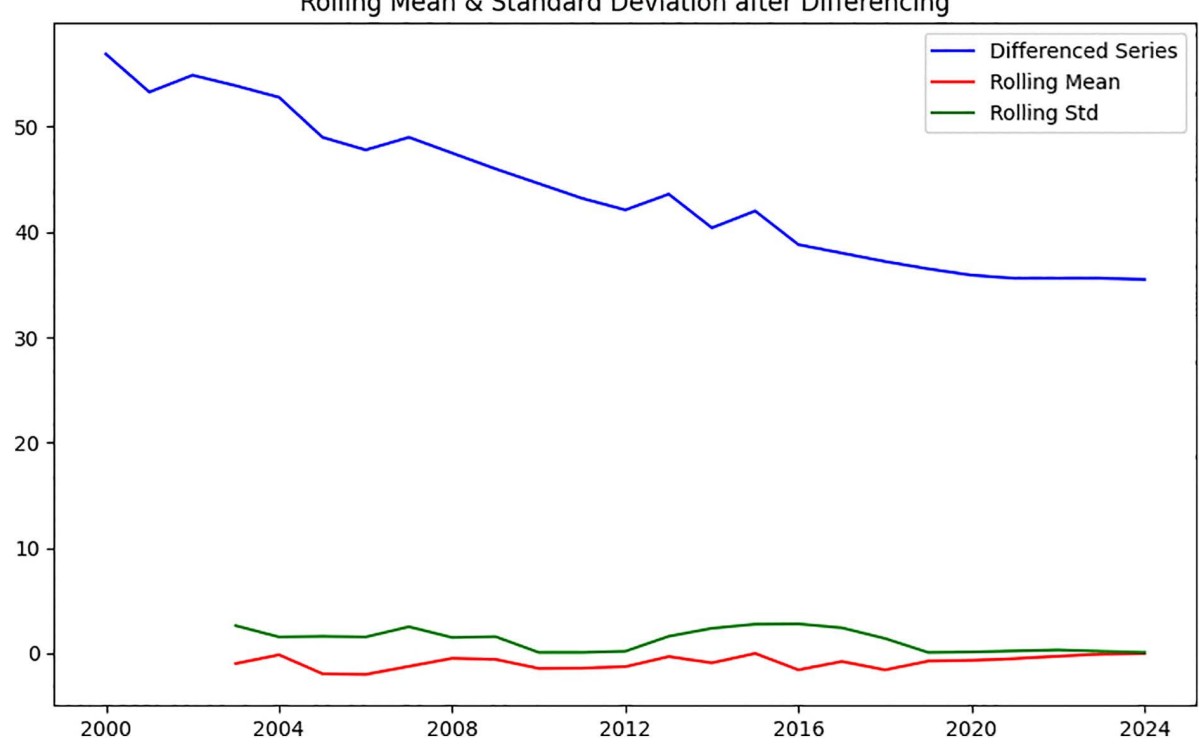

**Fig 3. Rolling mean and SE after differencing.**

its best performance with 4 units, a dropout rate of 0.2, L2 regularization of 0.001, **a** batch size of 8, 50 epochs, and a learning rate of 0.01. When evaluated using time-series cross-validation, this configuration yielded MAE of 4.52, MAPE of 11.27% and $R^2$ of 0.39%.

**Hybrid model.** Based on model performance evaluation, ETS (A,A,N) was identified as the best-performing statistical model. The ETS model produced residuals with a near-zero mean (0.0047) and standard deviation of 1.10, indicating only small fluctuations around the fitted trend. Analysis further showed no significant autocorrelation at lags 1–3 (0.078, 0.012, –0.230), confirming that the residuals essentially represented random noise and that the ETS model had already captured the predictable patterns in the series.

Despite the absence of residual patterns, the LSTM was trained using time-series cross-validation with hyperparameters optimized to minimize MAE (units = 4, dropout rate = 0.2, L2 regularization = 0.001, batch size = 8, epochs = 50, learning rate = 0.01). The resulting hybrid model achieved MAE = 1.69, RMSE = 1.82, and MAPE = 309.8%. In comparison, a naive predictor using the mean of the residuals achieved a lower MAE of 1.04, indicating that the hybrid ETS–LSTM model performed worse than a simple baseline.

These findings confirm that, although the hybrid approach was performed, adding an LSTM to the ETS residuals did not capture additional meaningful patterns and likely overfit the noise. Therefore, the ETS (A,A,N) model remains the most reliable forecasting model for under-five stunting prevalence (S3 Fig).

## Six-year forecast of under-five stunting in Ethiopia

Based on model evaluation, the ETS (A, A, N) model was selected as the most reliable for forecasting under-five stunting prevalence. Using this model, the prevalence was projected for the next six years. By 2029, the stunting prevalence is

projected at 32.35% (95% CI: 26.90–37.79), and by 2030, it is projected at 31.95% (95% CI: 25.98–37.91), indicating a gradual decline over the forecast period if current trends continue (Fig 4).

The six-year forecast of under-five stunting prevalence is summarized in Table 1, showing the projected values and their 95% confidence intervals (Table 1).

## Discussion

This study projected the prevalence of under-five stunting in Ethiopia from 2000 to 2030 using classical time series and machine learning models. Historical data showed a declining trend, with stunting prevalence decreasing from approximately 57% in 2000 to 35% in 2024. The observed decline in under-five stunting prevalence in Ethiopia from 2000 to 2024 can be attributed to multiple factors, including sustained public health interventions, expanded access to primary healthcare services, and multisectoral initiatives such as the Sekota Declaration, which foster collaboration across sectors addressing food security, sanitation, and education [32–35].

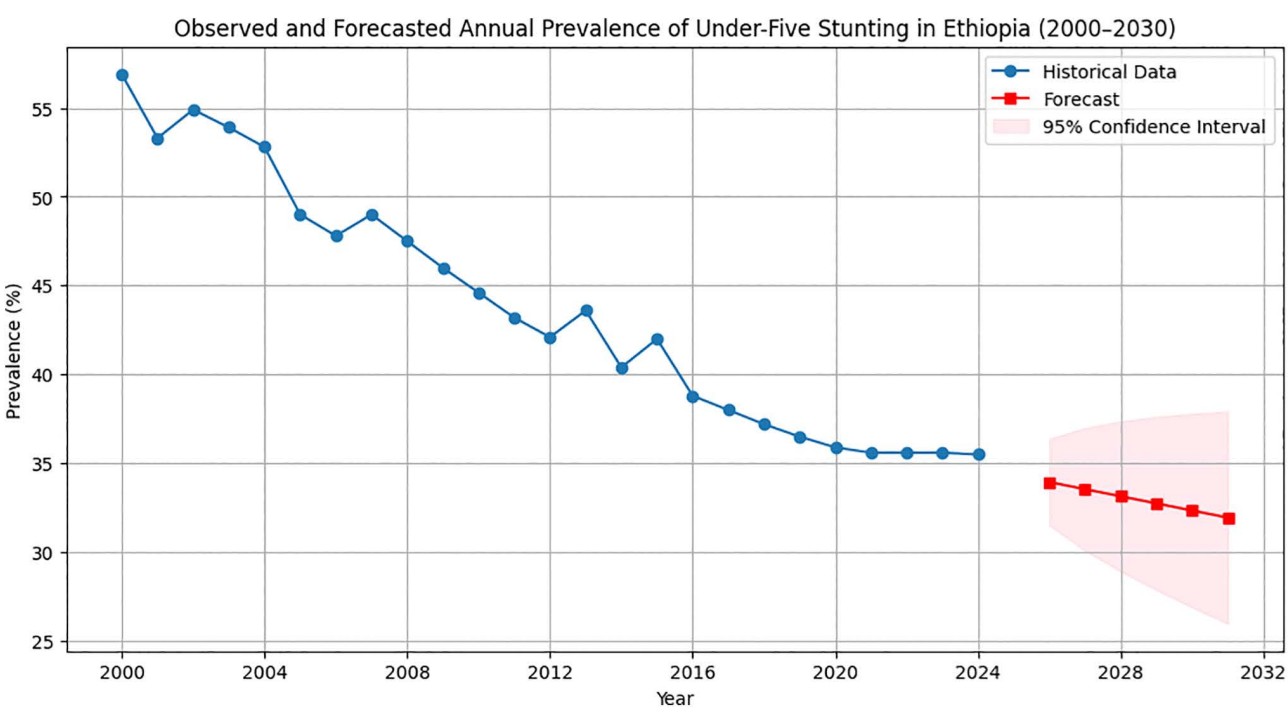

**Fig 4. Observed and forecasted Prevalence of under-five stunting in Ethiopia.**

**Table 1. Six-year forecast of under-five stunting prevalence.**

| Year | Forecasted Prevalence (%) | 95% Confidence Interval (%) |
|---|---|---|
| 2025 | 33.95 | 31.51–36.38 |
| 2026 | 33.55 | 30.11–36.99 |
| 2027 | 33.15 | 28.93–37.37 |
| 2028 | 32.75 | 27.88–37.62 |
| 2029 | 32.35 | 26.90–37.79 |
| 2030 | 31.95 | 25.98–37.91 |

Among the models evaluated, ETS was the most reliable, capturing the trend effectively (MAE = 1.09, MAPE = 2.71%, $R^2 = 0.9028$). This suggests that classical statistical methods can perform well when dealing with relatively small datasets, where deep learning or more complex machine learning models may risk overfitting or require larger training samples to achieve stable results. The superior performance of the ETS model in forecasting under-five stunting prevalence can be explained by several methodological factors. First, the relatively small sample size of the annual stunting series increases the risk of overfitting for complex machine learning and deep learning models, whereas ETS, with its parsimonious parameter structure, can produce stable and reliable forecasts [36] Second, ETS is particularly well suited to univariate forecasting tasks, as it relies solely on historical values and does not require additional explanatory covariates. In the absence of such covariates, machine learning and deep learning models may struggle to identify meaningful patterns, whereas ETS remains robust and interpretable [37]. Collectively, these factors highlight why ETS can outperform more complex models in settings with limited, smooth, and univariate health outcome data. This finding is also consistent with previous studies, which have demonstrated that exponential smoothing methods are often robust, interpretable, and computationally efficient for forecasting health-related outcomes with limited data [38,39].

The prevalence of under-five stunting is projected to be 32.35% (95% CI: 26.9–37.79) by 2029, which remains above Ethiopia's national target of 19% set by the Ministry of Health's National Nutrition Program [16]. By 2030, it is expected to decline further to 31.95% (95% CI: 25.98–37.91); however, this projected prevalence still exceeds the SDG 2.2 aim of ending all forms of malnutrition by 2030 [40].

Despite ongoing nutrition initiatives, several factors may hinder the achievement of these targets. Ethiopia has experienced decades of challenges stemming from drought, armed conflict, climate change, political and economic instability, the COVID-19 pandemic, and the recent surge in living costs these challenges, along with poor diet quality, unsafe complementary feeding practices, continue to exacerbate children's vulnerability to stunting and malnutrition [41–48]. Indicating that current interventions alone may be insufficient.

While these findings provide valuable insights into Ethiopia's progress toward national and global nutrition targets, it is important to interpret them in light of the study's strengths and limitations. A key strength is the use of 24 years of nationally representative data from the WHO Global Health Observatory, which provides a robust evidence base for examining long-term stunting trends. In addition, the application and comparison of multiple statistical and machine learning models enhanced the reliability of the projections. Benchmarking the results against Ethiopia's national nutrition targets and the global SDG 2.2 also ensured that the findings are directly relevant to both national policy and global commitments. However, the analysis was conducted using national-level annual data, which may obscure important regional and sub-population disparities in stunting prevalence. Furthermore, the dataset did not include socioeconomic or other contextual variables, limiting the ability to account for factors that could influence stunting trends.

## Conclusion and recommendation

This study projected the prevalence of under-five stunting in Ethiopia using 24 years of nationally representative data from the WHO Global Health Observatory. The findings show a gradual decline in stunting prevalence, with levels projected to reach 32.35% by 2029 and 31.95% by 2030. Although this indicates progress, the projected prevalence remains above Ethiopia's national target of 19% by 2029 and the global SDG 2.2 aim of ending all forms of malnutrition by 2030. The projections indicate that, despite ongoing nutrition initiatives, Ethiopia is not on track to achieve its targeted reduction in under-five stunting under current conditions.

We recommend conducting a national-level study to assess whether current nutrition programs are on track, identify regional or local discrepancies, and inform interventions tailored to specific community needs. This approach will help ensure strategies are effective, context-specific, and capable of supporting progress toward achieving stunting reduction targets.

## Supporting information

**S1 Fig. Autocorrelation and partial autocorrelation plots of under-five stunting prevalence in Ethiopia.**
(TIF)

**S2 Fig. STL (Seasonal-Trend Decomposition) of under-five stunting time series, showing observed, trend, seasonal, and residual components.**
(STIF)

**S3 Fig. Residual statistics of forecasting models for under-five stunting, illustrating model fit and unexplained variability.**
(TIF)

**S1 File. Who Stunting data ethiopia.**
(CSV)

## Author contributions

**Conceptualization:** Rewina Tilahun Gessese.

**Formal analysis:** Rewina Tilahun Gessese, Tigist Kifle Tsegaw.

**Methodology:** Rewina Tilahun Gessese, Zinabu Bekele Tadese, Eliyas Addisu Taye.

**Software:** Rewina Tilahun Gessese, Fetlework Gubena Arage.

**Supervision:** Rewina Tilahun Gessese.

**Validation:** Rewina Tilahun Gessese, Jenberu Mekurianew Kelkay, Eyob Akalewold Alemu.

**Visualization:** Rewina Tilahun Gessese.

**Writing – original draft:** Rewina Tilahun Gessese.

**Writing – review & editing:** Jenberu Mekurianew Kelkay, Fetlework Gubena Arage, Tigist Kifle Tsegaw, Zinabu Bekele Tadese, Meron Asmamaw Alemayehu, Eliyas Addisu Taye, Eyob Akalewold Alemu.

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
