## [Decision Letter · Decision Letter 0]

10 Feb 2026

Dear Dr. Gessese,

Thank you for submitting your manuscript to PLOS ONE. After careful consideration, we feel that it has merit but does not fully meet PLOS ONE’s publication criteria as it currently stands. Therefore, we invite you to submit a revised version of the manuscript that addresses the points raised during the review process.

We look forward to receiving your revised manuscript.

Kind regards,

Zaher Mundher Yaseen

Academic Editor

PLOS One

Journal Requirements:

2. Please note that your Data Availability Statement is currently missing the repository name and/or the DOI/accession number of each dataset OR a direct link to access each database. If your manuscript is accepted for publication, you will be asked to provide these details on a very short timeline. We therefore suggest that you provide this information now, though we will not hold up the peer review process if you are unable.

Reviewers' comments:

Reviewer's Responses to Questions

**Comments to the Author**

1. Is the manuscript technically sound, and do the data support the conclusions?

Reviewer #1: Yes

Reviewer #2: Yes

2. Has the statistical analysis been performed appropriately and rigorously?

Reviewer #1: Yes

Reviewer #2: Yes

3. Have the authors made all data underlying the findings in their manuscript fully available?

Reviewer #1: Yes

Reviewer #2: Yes

4. Is the manuscript presented in an intelligible fashion and written in standard English?

Reviewer #1: Yes

Reviewer #2: Yes

Reviewer #1: The study addresses an important public health issue and demonstrates a sound methodological approach by using reliable secondary data and comparing classical and machine learning time series models; however, the manuscript would benefit from clearer reporting of time series stationarity diagnostics and a more detailed description of hyperparameter tuning for the machine learning models. In addition, the exclusive reliance on annual data and the absence of explanatory socioeconomic or contextual variables limit the analytical depth of the forecasts. The superior performance of the ETS model compared to machine learning approaches warrants further methodological discussion, and the conclusions could be strengthened by more specific, results-driven policy recommendations. These limitations are primarily developmental and do not detract from the overall scientific merit of the study.

Reviewer #2: Well-organized paper. The motivation behind this paper is good. I didn't see any major issue regarding the methods or the data analysis. Data analysis was reasonably thorough. The authors performed necessary preprocessing. Both the statistical and the deep learning methods used in this work are known to perform well for time series data. The authors also took care of the issues, e.g., overfitting, that arise when one trains a machine learning model, and provided the details. Presentation of the results was good.

Minor editing issues, e.g., space between a wort and punctuation, etc., exist. Another proof-read should be conducted to correct these issues.

.

Reviewer #1: No

Reviewer #2: No

---

## [Author Response · Author response to Decision Letter 1]

18 Feb 2026

Comment / Requirement Response / Revision

Reviewer #1: The manuscript would benefit from clearer reporting of time series stationarity diagnostics. We have added a detailed description of the stationarity assessment in the Methods section. Visual inspection of rolling statistics (mean and standard deviation) and formal Augmented Dickey-Fuller (ADF) tests are now reported. The original series was non-stationary (ADF p = 0.62), and stationarity was achieved after first differencing (ADF p < 0.001). Figures illustrating these checks are included as Figure 2 and 3.

Reviewer #1: A more detailed description of hyperparameter tuning for the machine learning models is needed. The Methods section has been expanded. For MLP, hyperparameters (hidden layers, activation functions, learning rates, batch sizes, L2 regularization, max iterations) were tuned via randomized search with early stopping. For LSTM, hyperparameters (units, dropout rate, learning rate, batch size, number of epochs) were tuned via manual random search with early stopping and learning rate reduction callbacks. The final selected models and rationale for shallow architectures are explicitly described.

Reviewer #1: Exclusive reliance on annual data and absence of explanatory socioeconomic/contextual variables limit analytical depth. We acknowledge this limitation. The Discussion now explicitly notes that forecasts are based on national-level annual data without socioeconomic or contextual covariates, which may limit the ability to account for drivers of stunting trends. Additionally, national-level data may obscure important regional or sub-population disparities.

Reviewer #1: The superior performance of the ETS model compared to machine learning approaches warrants further methodological discussion. We added a detailed explanation of why ETS outperformed the machine learning models: 1) Small sample size increases overfitting risk for complex models, whereas ETS’s parsimonious structure provides stable forecasts. 2) ETS is well-suited for univariate forecasting and does not require covariates. 3) Smooth trends in the annual stunting series favor classical exponential smoothing methods. Relevant references have been included.

Reviewer #1: Conclusions could be strengthened by more specific, results-driven policy recommendations. We revised the Conclusion section to include results-driven recommendations based on our projections: the projected under-five stunting prevalence (~32% by 2030) exceeds Ethiopia’s national target (19%) and the SDG 2.2 goal. We recommend focused research to identify factors driving high stunting prevalence and evidence-based, context-specific interventions to support achievement of national and global targets.

Reviewer #2: Well-organized paper; motivation is good; methods and analysis are thorough; minor editing issues (spacing, punctuation, etc.). We thank the reviewer for the positive feedback. We have conducted an additional thorough proof-read of the manuscript to correct spacing, punctuation, and other typographical errors.

Editor: Ensure manuscript meets PLOS ONE style requirements, including file naming. The manuscript has been reformatted according to the PLOS ONE style templates for both main body and title/authors/affiliations. All file names now comply with the journal guidelines.

Editor: Provide repository name and/or DOI/accession number or direct link in Data Availability Statement. The Data Availability Statement has been updated to include the WHO Global Health Observatory repository and the direct link for the under-five stunting prevalence data: https://data.who.int/indicators/i/5F8A486?utm_source

Editor: Include captions for Supporting Information files and update in-text citations. Captions for all Supporting Information files have been added at the end of the manuscript, and all in-text citations have been updated accordingly.

Editor: Review reviewer-recommended citations for relevance. No additional citations were recommended by the reviewers.

Editor: Review and ensure references are complete and correct; remove or indicate retracted articles. The reference list has been thoroughly reviewed. Any retracted articles have been removed or replaced, and all references are complete and formatted according to PLOS ONE guidelines.

---

## [Decision Letter · Decision Letter 1]

2 Mar 2026

Forecasting Under-Five Stunting in Ethiopia Using Classical and Machine Learning Time Series Models

PONE-D-25-57738R1

Dear Dr. Gessese,

We’re pleased to inform you that your manuscript has been judged scientifically suitable for publication and will be formally accepted for publication once it meets all outstanding technical requirements.

Kind regards,

Zaher Mundher Yaseen

Academic Editor

PLOS One

Additional Editor Comments (optional):

Reviewers' comments:

Reviewer's Responses to Questions

**Comments to the Author**

Reviewer #1: All comments have been addressed

Reviewer #2: All comments have been addressed

2. Is the manuscript technically sound, and do the data support the conclusions?

Reviewer #1: Yes

Reviewer #2: Yes

3. Has the statistical analysis been performed appropriately and rigorously?

Reviewer #1: Yes

Reviewer #2: Yes

4. Have the authors made all data underlying the findings in their manuscript fully available?

Reviewer #1: Yes

Reviewer #2: Yes

5. Is the manuscript presented in an intelligible fashion and written in standard English?

Reviewer #1: Yes

Reviewer #2: Yes

Reviewer #1: All comments have been addressed. No further correction needed at this stage. Congratulations for the authors

Reviewer #2: I didn't have any major concerns. The minor issues I mentioned have been addressed. This is an well-motivated and well-organized paper.

.

Reviewer #1: No

Reviewer #2: No

---

## [Editor Report · Acceptance letter]

PONE-D-25-57738R1

PLOS One

Dear Dr. Gessese,

I'm pleased to inform you that your manuscript has been deemed suitable for publication in PLOS One. Congratulations! Your manuscript is now being handed over to our production team.

Kind regards,

on behalf of

Dr. Zaher Mundher Yaseen

Academic Editor

PLOS One